# Comparative Efficacy of Pyrethroid-Based Paints against Turkestan Cockroaches

**DOI:** 10.3390/insects15030171

**Published:** 2024-03-03

**Authors:** Miguel Salazar, John L. Agnew, Alvaro Romero

**Affiliations:** Department of Entomology, Plant Pathology and Weed Science, New Mexico State University, Las Cruces, NM 88003, USAagnewj@nmsu.edu (J.L.A.)

**Keywords:** peridomestic cockroach, *Periplaneta lateralis*, nymphs, invasive urban pest, insecticide paint, alphacypermethrin, deltamethrin, transfluthrin, substrate type, exposure time

## Abstract

**Simple Summary:**

Turkestan cockroaches are primarily outdoor nuisance pests in many parts of the world. The application of pesticide sprays in and around areas where cockroaches aggregate is a common practice in trying to reduce their populations. Intensive pesticide treatments can lead to pesticide runoff into water sources, such as rivers, lakes, streams, and dwells. Insecticide-based paints offer an alternative option for long-term efficacy in controlling cockroaches with diminished environmental impacts. In this study, pyrethroid-based paints applied to concrete, metal, and PVC—common surfaces where cockroaches reside—were demonstrated to kill and repel Turkestan cockroach nymphs. The pyrethroid alphacypermethrin produced the highest mortality most rapidly. The lethal and repellency effects of these paints may help prevent the establishment of Turkestan cockroaches around buildings. These results support the use of insecticide-paint technologies in areas where Turkestan cockroaches are a dominant pest.

**Abstract:**

The Turkestan cockroach, *Periplaneta lateralis* (Walker), is an invasive urban pest prevalent in dry areas of the southwestern United States. Treatment with liquid spray formulations containing insecticides is the most conventional method to decrease Turkestan cockroach population abundance around buildings. Intensive application of insecticide treatments near natural environments has prompted concerns regarding the impacts on non-target aquatic and terrestrial ecosystems. Technologies embedding insecticides in a paint matrix have successfully been used for the long-term reduction in disease-vector populations in tropical areas. Here, we evaluated the potential effectiveness of three pyrethroid-based paints against Turkestan cockroach nymphs on common surfaces inhabited by this species. Turkestan cockroaches continuously exposed for 1 h to 1-month aged alphacypermethrin and deltamethrin paints applied to concrete, metal, or PVC caused moderate to high mortality. Evaluations using choice boxes indicated that deltamethrin and transfluthrin paints had combined lethal and repellent effects on cockroaches. Alphacypermethrin also caused repellency and killed cockroaches rapidly. We discuss the implications of these findings on cockroach control practices.

## 1. Introduction

The Turkestan cockroach, *Periplaneta lateralis* (Walker) (Blattodea: Blattidae), is a peridomestic invasive species found in many parts of the world [1,2,3,4]. Turkestan cockroaches are well adapted to the dry conditions of the southwestern United States, and they have been reported to have spread to other regions of the United States (https://www.inaturalist.org/observations?taxon_id=1473952, accessed on 1 March 2024). This expansion of Turkestan cockroaches has been attributed partly to their biological and reproductive advantages over *Blatta orientalis*—a species occupying similar habitats [5,6]. In urban settings, these cockroaches are found primarily in areas around homes and other structures (e.g., irrigation boxes, water meter boxes, gardens, and crawlspaces), but they can access indoor spaces through holes, cracks, and door gaps, thus leading to the infestation of attics, wall voids, and sometimes kitchens and bathrooms [3,7,8]. Turkestan cockroach infestations have also been detected in sewer systems and around animal barns [3]. For the management of this cockroach, programs that include outdoor application of baits and exclusion have demonstrated to reduce populations within a month [9]. Although common insecticide formulations have shown to be effective against Turkestan cockroaches in laboratory studies [10], their lasting insecticidal effects are often limited by the rapid degradation of active ingredients (AIs) by ultraviolet light [11] and the alkalinity of concrete and plaster substrates [12], thus resulting in residual activity persisting for no more than a few weeks. In addition, insecticide treatments are applied outdoors, and environmental concerns have been raised because pyrethroid insecticides have been detected in urban creeks and natural areas [13], as well as in non-target aquatic and terrestrial ecosystems [14] and have been found to leach into groundwater-based drinking water resources [15].

The use of surface coatings with insecticides is a well-known method to improve the performance of residual insecticides [16,17,18]. Lacquers are common additives to insecticides used to prolong their activity against cockroaches [12]. Enhanced residual toxicity of lacquer-formulated organophosphates [19], organochlorine [16], and pyrethroids [17] was demonstrated against German cockroaches. Micro-encapsulation of active ingredients in a polymeric matrix is another development of a controlled released delivery system of insecticides reported to have residual efficacy against cockroaches and some crawling insects [12,20]. The novel advancement of paint formulation by imbedding microencapsulated insecticide into latex paint offers a prolonged residual effect as compared to earlier insecticide paint (IP) products [21,22]. In these paints, the AIs slowly migrate to the substrate surface as the paint dries, meaning the insecticide is usable for a prolonged time. In addition, they are ready-to-use products that can be applied to surfaces by spraying or brushing [21]. The insecticidal activity of these IP formulations has been found to be effective against mosquitoes for up to one year after application [23] and up to 32 months against triatomine kissing bugs [24]. Contact or proximity of insects to IPs can cause broad toxicological effects, including irritancy, repellency, and mortality [25]. Application of IPs in interior and exterior areas has been used to protect against vector-borne diseases including Chagas, leishmaniasis, lymphatic filariasis, dengue, and chikungunya [21]. IPs have been suggested as the preferred option for insects that establish and colonize exteriors, such as sand flies, a vector of leishmaniasis [21,23,24,25].

The purpose of this study was to determine the potential of three paints containing pyrethroids alphacypermethrin, deltamethrin, or transfluthrin for the management of Turkestan cockroaches.

## 2. Materials and Methods

### 2.1. Cockroaches

Cockroaches were obtained from a colony maintained at 23–26 °C, a relative humidity of 30–50%, and a photoperiod of 12:12 (L:D) h. This colony was originally established from samples collected at the New Mexico State University (NMSU) feed mill facility in Las Cruces in 2017. Colonies were maintained in plastic containers provided with egg crates for shelter, water, and food. Cockroach food consists of a mix in equal proportion of rabbit feed, dog feed, cat feed, and sweetened puffed corn cereal. We used nymphs because they were the most abundant life stage of the colony and they are also the most common foraging life stages (A.R.; pers. comm.). A previous evaluation showed that the colony used in this study was susceptible to various insecticide formulations [10].

### 2.2. Insecticide Paints

Three paints were evaluated: INESFLY 7A DELTA (0.25% deltamethrin), INESFLY VESTA paint (0.5% transfluthrin), and A-cypermethrin IP (0.7% A-cypermethrin) (Inesfly Corporation, Valencia, Spain). The IPs were applied following application rates recommended by the manufacturers: 175 g/m^2^ on concrete and polyvinyl chloride (PVC), and 140 g/m^2^ on metal. Concrete received two paint layers, according to the manufacturers’ instructions.

### 2.3. Forced Exposure Bioassays

This study used three substrates: one porous material comprising concrete slabs (pewter square concrete step stone, 30.48 cm × 30.48 cm × 3.81 cm, Home Depot), and two non-porous materials comprising solid steel sheet metal (gauge weldable sheet, 30.48 cm × 30.48 cm, 22-gauge, Home Depot) and PVC caps (8 cm diameter Knock-Out PVC DWV Test Caps, Oatey). For the concrete and steel, 8 cm diameter circles were demarcated with pencil to match the diameters of the PVC caps, and each substrate was painted (2 1/2 in. Pro Nylon/Polyester Thin Angle Sash Brush) with each of the three paint treatments, whereas the fourth circle was left untreated (control) (Figure 1).

To each concrete circle or PVC cap, 0.88 g of each IP was applied; metal circles received 0.7 g. The application amount was doubled for concrete circles (1.76 g, according to the manufacturers’ recommendations); the second paint layer was applied two hours after drying. The total amounts of active ingredient (mg) on each substrate are displayed in Table 1.

The painted substrates were allowed to dry for 7 days (according to the manufacturers’ recommendations) and were then used immediately (denoted “fresh” hereafter). Another set of 7-day dried substrates was aged for 1 month at room temperature before use (denoted “1 month old” hereafter). For each of the three IP treatments on a single substrate, ten cockroach nymphs were forcibly exposed and this experimental setup was replicated five times. This was carried out for both 15 min and 1 h forced exposures resulting in 1200 total Turkestan nymphs being used in this study. Since Turkestan cockroaches cannot climb smooth vertical surfaces (A.R., pers. comm.), PVC rings (8 cm diameter, 8 cm height) were placed on the painted areas, ensuring continuous exposure and preventing escape (Figure 1).

After exposure, the insects were then moved to clean areas (deli cups, 473 mL, Round Deli Containers, ProPak, Houston, TX, USA), provided with food and water, and maintained in a room with a controlled environment (24–26 °C and 40 ± 10%). Mortality was monitored daily until day 14. Cockroaches not responding by contact with entomological forceps were classified as dead.

### 2.4. Choice-Box Experiments

Cockroach repellency and mortality were determined with modified Ebeling choice boxes [26] (Figure 2). The box was square (30.48 cm side) with a bottom made of PVC (Sibe-R Plastic Supply PVC, Ocala, FL, USA) and walls made of foam (3 mm thick handicraft white foam board, Hobby Lobby). The box was divided into two equal compartments by a vertical partition. A rectangular hole in the partition (2.5 cm height, 5.0 cm base) allowed the cockroaches to move freely between the compartments. Food and water were placed in the lighted compartment in the choice tests. In treated boxes, the flooring under the dark side was painted with the IPs (area treated 464.5 cm^2^, application rate 175 g/m^2^, 8.2 g), whereas the control boxes remained unpainted under the dark side (Figure 2). Painted substrates were allowed to dry for 7 days before use. The lighted compartment was covered with mesh (500 µm plankton netting, Bioquip Products, Rancho Dominguez, CA, USA) fixed to the foam board with pins. The choice boxes were exposed to a photoperiod of 12:12 (L:D) h.

Twenty Turkestan cockroach nymphs in the third instar were released into the untreated compartment within 2 h into the scotophase and were allowed to enter the treated compartment through removal of the cork ~12 h later. Shop lights fitted with white fluorescent bulbs were placed above the choice boxes. The mesh covering half of the choice box was exposed to a light intensity of 300–350 lux (INS Digital Lux Meter, Markson Scientific, Phoenix, AZ, USA). Bioassays were conducted under ambient temperature (24–26 °C) and relative humidity (40 ± 10%). The number of live and dead cockroaches in each compartment at 3–4 h into the photophase was recorded daily for 14 d. Repellency was defined as the mean percentage of live cockroaches present in the light compartment.

### 2.5. Statistical Analysis

Mortality data from forced exposure and Ebeling choice experiments were analyzed with a Kaplan–Meier survivorship curve [27] generated in SPSS 22 (IBM 2013, Corporation, Chicago, IL, USA). Mortality data in lighted and dark areas were added before analysis. The log-rank test was used to identify differences among treatment groups, and pairwise comparisons were conducted across treatments. Daily repellency was analyzed with repeated measure ANOVA. The Bonferroni method was used for pairwise comparison (*p* < 0.05) (SPSS 22, IBM 2013).

## 3. Results

### 3.1. Forced Exposure

The effectiveness of IPs varied according to the exposure time and surfaces (Figure 3). In addition, 15 min exposures to fresh paints on concrete, metal, or PVC caused minimal mortality (≥86% survivorship) (Figure 3), which did not significantly differ from the mortality observed after exposure to the unpainted control substrates (*p* > 0.05). For all substrates with 0.25% deltamethrin or 0.7% alphacypermethrin paints, the groups exposed for 1 h to 1-month aged paint had greater mortality than the groups with 15 min or 1 h exposure times to fresh paint, as well as the control group (Figure 3, Appendix A). The highest mortality was observed in cockroaches exposed to treated metal for 1 h to 1-month aged alphacypermethrin or deltamethrin paint (mortality at day 14 = 82% and 54%, respectively) (Appendix A). On concrete and PVC, the mortality of cockroaches exposed to 1-month aged alphacypermethrin or deltamethrin paints did not significantly differ (*p* > 0.05). However, 1-month aged alphacypermethrin paints on metal were significantly more effective against cockroaches than deltamethrin paints aged for the same time (χ^2^ = 8.78, *p* = 0.03). Transfluthrin applied on concrete, metal, or PVC did not cause significant cockroach mortality, even when the insects were exposed to 1-month aged paints for 1 h (*p* > 0.05) (Figure 3) (Appendix A).

### 3.2. Choice Box Experiments

#### 3.2.1. Repellency

During all 14 days of the experiment, the transfluthrin and deltamethrin-treated choice boxes, compared with alphacypermethrin and the control boxes, showed significant repellent activity against Turkestan cockroach nymphs (F = 15.57, df = 3, *p* < 0.04) (Figure 4). On day 1, all IPs significantly repelled more cockroaches (range 50–80% repellency) than the untreated control (0% repellency) (Figure 4). Since day 2, cockroaches in transfluthrin (~60–70%) and deltamethrin boxes (~40–60%) remained on the untreated area until the end of the experiment (Figure 4), and no significant differences were detected between these groups (*p* > 0.05) (Figure 4).

#### 3.2.2. Mortality

Overall, alphacypermethrin paints killed significantly more cockroaches than the other IPs (χ^2^ = 229.27, df = 83, *p* < 0.001) (Figure 5). By day 2, ~95% of cockroaches were dead in the alphacypermethrin-treated boxes, whereas ~37% and ~28% had died in boxes with deltamethrin and transfluthrin. By the end of the experiment, the cockroach mortality in the deltamethrin and transfluthrin boxes was ~58% and ~42%, respectively (Figure 5).

## 4. Discussion

We investigated the toxicity and repellency of pyrethroid-based paints to determine their potential for Turkestan cockroach control. Initial evaluations in forced exposure assays indicated that mortality varied according to the IP, exposure times and aging time of painted surfaces. Overall, IPs in forced exposure assays caused a broad range of cockroach mortality. Deltamethrin and alphacypermethrin caused moderate to high mortality, whereas transfluthrin did not cause mortality. The mortality observed in alphacypermethrin and deltamethrin provided evidence that the AI was available on the painted surfaces. Overall, however, mortality was lower than that observed in a recent study with the same cockroach population exposed to surfaces treated with a pyrethroid–based encapsulated suspension formulation [10]. In that study, exposures of Turkestan cockroach nymphs for 5 min on concrete, tile, or wood to dry residues of 0.06% lambda cyhalothrin (at least four times lower than the pyrethroid concentrations contained in the paints) killed 100% of individuals within 4 d [10]. Penetration of insecticides through the insect’s cuticle is highly dependent on the solvents in the formulation [28]. Therefore, solvents in the encapsulated concentration formulation that enhance the penetration of the AIs through the cuticle might explain the differences in cockroach mortality. Likewise, the low lipid solubility of water-based paints might have affected the effective acquisition of AIs by Turkestan cockroaches from the substrates used in this study. The lack of efficacy of transfluthrin paints in our study differs from reports of their high and long efficacy against *Anopheles gambiae* mosquitoes exposed to the same formulation for similar times [29]. Different penetration rates of transfluthrin through the cuticle might explain the differences in mortality observed between these two insects. In addition, the greater water solubility of transfluthrin than other pyrethroids [30] might have affected the translocation of the insecticide through the cuticle.

IP-treated surfaces were used after a 7-day drying period, during which the AI was expected to migrate to the surface, as indicated by the manufacturer. However, 15 min continuous exposure of cockroaches to dried surfaces was not sufficient for killing. The mortality slightly increased in cockroaches exposed for 1 h on concrete and metal surfaces with fresh alphacypermethrin. Longer exposure times to treated surfaces have been found to lead to substantially higher mortality in German cockroaches because of the greater uptake of insecticides [12]. In addition, we observed that Turkestan cockroaches increased their locomotor activity after contact with alphacypermethrin-treated surfaces; this irritancy is a known effect of alphacypermethrin [31] and might potentially have accelerated exposure to lethal doses of insecticide [32]. Also, vapors released from the insecticide-painted surfaces could have also contributed to cockroach mortality [33].

Aging the alphacypermethrin-painted surfaces for 1 month on all substrates further increased cockroach mortality, thus indicating greater bioavailability of the AI on the surface over time. Similar results were observed in German cockroaches exposed to surfaces painted with lacquer-formulated fenvalerate aged for 90 days [17] and this was attributed to the migration of the AI to the surface. Deltamethrin did not cause mortality as fresh paint. The limited toxicity of fresh deltamethrin paints might be associated with slower migration to the surface than that of alphacypermethrin, or to its modification within the paint matrix. Previous studies have reported that deltamethrin is sensitive to the alkaline environment in the paint and subsequently degradates into by-products, thus decreasing the concentration of α-DLM, the known active form of deltamethrin [34]. However, the higher mortalities observed after exposure to 1-month aged deltamethrin paints contradict this argument and suggested a greater bioavailability of deltametrin on the surface over time as the paint aged.

The potential performance of IPs against Turkestan cockroaches was determined through analysis of repellency and mortality values estimated from Ebeling box experiments. Experiments in these boxes indicate the positional responses of Turkestan cockroaches offered insecticide-treated areas that are normally attractive, such as dark voids and harborages [26]. We initially evaluated repellency by quantifying the number of cockroaches found alive in the lighted area of the choice box over 14 days (Figure 4). On day 1, between 50 and 80% of Turkestan cockroaches were positioned away from the insecticide-painted areas. From day 2 until the end of the evaluation period, the proportion of cockroaches avoiding the transfluthrin and deltamethrin areas remained steady. Insects have behavioral avoidance responses that may decrease their exposure to insecticides [35]. Avoidance to pyrethroids has been documented in other crawling household pests such as bed bugs [36], ants [37], and kissing bugs [38]. In our evaluations, cockroaches entered the treated areas within the first hour, and insecticide contact provoked their direct movement away from the treated areas. Pyrethroids with high vapor pressures, such as transfluthrin, are used as spatial repellents against mosquitoes [39]. Less volatile pyrethroids, such as alphacypermethrin and deltamethrin, are known for their excitatory-repellent activity upon direct contact with treated surfaces [36,40]. The repellent responses of cockroaches to deltamethrin or transfluthrin-painted areas can have beneficial or adverse effects, depending on the circumstances. For example, cockroaches that survive after encountering insecticide-painted areas around buildings, or indoor areas such as attics or basements, would be discouraged from colonizing such areas. However, little benefit would be expected when individuals encounter sublethal doses, or incomplete IP coverage, which would allow insects to move into insecticide-free areas. Under this scenario, the combined use of insecticide paints with low impact insecticides such as dry baits, and exclusion, could be effective for the sustainable control of Turkestan cockroaches, particularly in protected areas around buildings [8,9].

A key aspect for determining the potential of insecticide-based paints for cockroach control is the evaluation of their effectiveness on surfaces found around buildings where cockroaches reside and aggregate. The physical and chemical characteristics of substrates affect the bioavailabity of insecticide on the surfaces. Insecticide residual sprays are more short-lived on porous (e.g., concrete, mud) than non-porous (e.g., vinyl siding or tile) surfaces [12]. Our results in forced exposure assays indicated that the IPs performed similarly on concrete, metal, and PVC. Previous studies have reported diminished effectiveness of IPs applied to concrete surfaces [23]. To counteract the surface effects for mosquito control, Mosquiera et al. [23] covered porous surfaces with regular paint before applying the IP, or applied two layers of IP. Given the great variety of building materials, the benefits of the use of IPs in cockroach management programs will largely depend on knowledge regarding the performance of paints on predominant building materials where infestations occur.

Prolonged lasting effects on surfaces are a desired feature of residual insecticides for pest control, particularly in areas continually challenged by infestations [11]. In our study, we did not test the efficacy of the paints beyond 1 month and future studies should evaluate the efficacy of these materials for longer periods. However, field and laboratory studies have demonstrated the efficacy of paints for at least 1 year against disease vectors [23,40]—effects notably superior to those of traditional insecticide residual sprays. The long-term efficacy of insecticide residues is also affected by rainfall, photodegration, temperature, and moisture [12], and the evaluation of these effects is essential when paints are to be applied to outdoor, non-protected spaces. Currently, limited information is available regarding the effects of environmental factors on the efficacy of insecticide-based paints. By evaluating the potential washoff of a transfluthrin-based IP through water immersion tests simulating a rain event, the minimum mass loss of transfluthrin has been estimated to range between 0.007 and 0.013% from the original paint concentration [41]. This loss of transfluthrin is many orders of magnitude lower than those reported for wettable powder and emulsifiable concentrate pyrethroid residual sprays (5.4 and 14.1%, respectively) [42]. Minimal mass loss of AI from pyrethroid-based paint could translate to longer residual insecticidal effects as well as a decreased insecticide washoff of the AI, which might reduce the effects on non-target organisms. Insecticide-based paints might impact beneficial insects resting or nesting in areas where these paints are intended to be used. Where there are concerns about this impact, paints should be restricted to areas less likely to be occupied by beneficial insects such as crawl spaces and attics.

The development of insecticide resistance in *Periplaneta* spp. is a concern with the use of pyrethroid-based paints [43]. Outdoor painted applications are subject to a variety of physical surface conditions and abiotic factors that can decrease insecticide efficacy over time [41]. Therefore, continuous exposure to sublethal doses of insecticides could select resistant individuals [44]. Studies on the efficacy and shelf life of insecticide-treated paints on surfaces would help determine reapplication times and the selection of active ingredients to rotate among these materials. The incorporation of different classes of insecticides (e.g., neonicotinoids, organophosphates, IGRs, and pyrroles), and synergists, such as piperonyl butoxide in IPs, is a strategy that may help delay insecticide resistance in cockroach populations [25].

## 5. Conclusions

This study demonstrated that insecticide-based paints have potential in the management of Turkestan cockroaches in microhabitats around buildings or areas unlikely to be hit by direct rainfall such as covered water meter boxes, drainage pipes, crawlspaces, and attics. Paints containing alphacypermethrin, deltamethrin, or transfluthrin had both mortality and repellency effects that might affect Turkestan cockroach population abundance. Among these AIs, alphacypermethrin might rapidly decrease cockroach population abundance indoors and around buildings. The potential benefits of extended residual effects on various substrates, and lower environmental impact, may be favorable in terms of the registration and approval processes of regulatory agencies, as well as consumer acceptability for Turkestan cockroach control.

## Figures and Tables

**Figure 1 insects-15-00171-f001:**
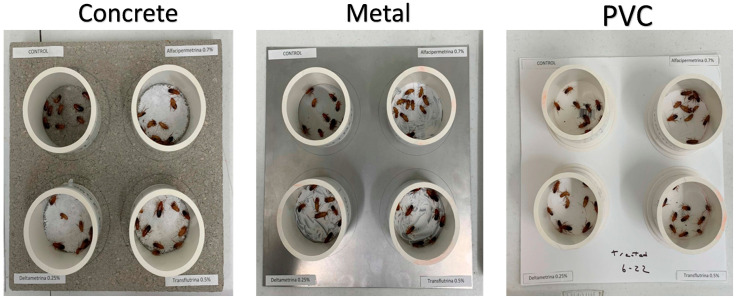
Groups of 10 cockroach nymphs were forcibly exposed for 15 min or 1 h to surfaces treated with insecticide-based paints. PVC rings prevented cockroach escape.

**Figure 2 insects-15-00171-f002:**
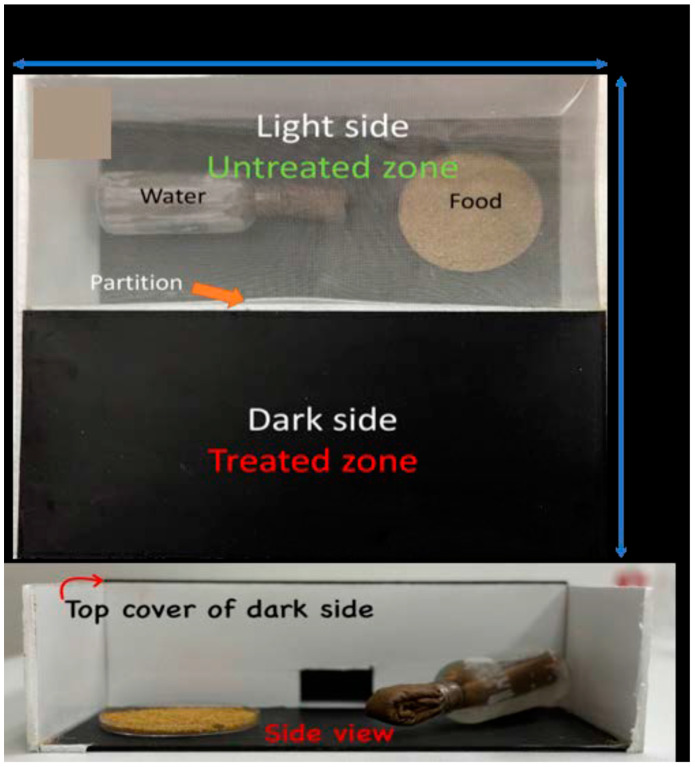
Modified Ebeling choice box to test the performance of insecticide-based paints. The flooring of dark areas received 20.5 mg of deltamethrin, 41 mg of transfluthrin, or 57.4 mg of alphacypermethrin. Cockroaches were released in the lighted area and allowed to move freely through a window located in the lower part of the dividing wall. Cockroaches were continuously exposed to the treated bottoms of the dark halves because they are unable to climb smooth vertical surfaces.

**Figure 3 insects-15-00171-f003:**
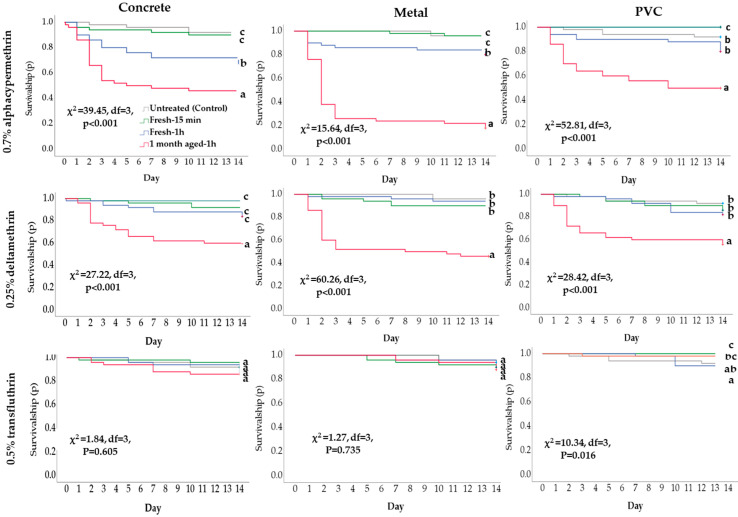
Kaplan–Meier survival analyses of Turkestan cockroach nymphs exposed for 15 min or 1 h to fresh or 1-month aged insecticide-based paints on various substrates. Different letters indicate a significant difference among treatments, according to the log-rank test (*p* < 0.05).

**Figure 4 insects-15-00171-f004:**
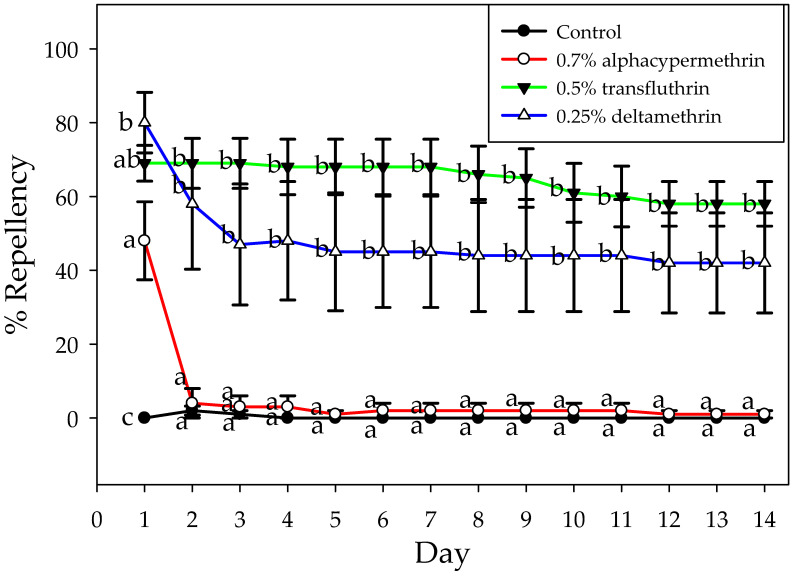
Repellency activity (mean ± SE) of insecticide-based paints against Turkestan cockroach nymphs in Ebeling choice boxes. Different letters indicate significant differences (one-way repeated measure ANOVA; adjustment for multiple comparisons: Bonferroni; *p* < 0.05).

**Figure 5 insects-15-00171-f005:**
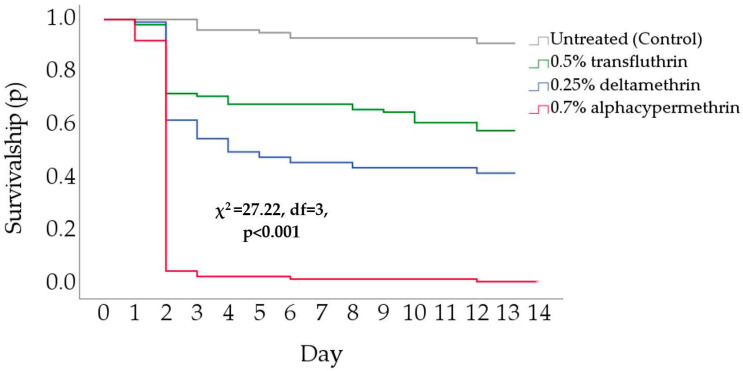
Kaplan–Meier survival analyses of Turkestan cockroach nymphs in Ebeling choice boxes. Different letters indicate a significant difference across treatments, according to the log-rank test (*p* < 0.05).

**Table 1 insects-15-00171-t001:** Amounts of insecticide paints applied to various substrates.

Substrate	Treated Area (cm^2^)	Total Paint (g)	Insecticide	mg Active Ingredient/Treated Area
PVC	50.2	0.88	0.25% deltamethrin	2.2
			0.5% transfluthrin	4.4
			0.7% alphacypermethrin	6.16
Metal	50.2	0.7	0.25% deltamethrin	1.75
			0.5% transfluthrin	3.5
			0.7% alphacypermethrin	4.9
Concrete	50.2	1.76	0.25% deltamethrin	4.4
			0.5% transfluthrin	8.8
			0.7% alphacypermethrin	12.3

## Data Availability

Datasets generated for this study can be obtained from the corresponding author.

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
