# Peer review of "Comparative Efficacy of Pyrethroid-Based Paints against Turkestan Cockroaches"

_insects, 2024, doi:10.3390/insects15030171_

Round 1

Reviewer 1 Report

Comments and Suggestions for Authors

The evaluated work 'Comparative Efficacy of Pyrethroid-Based Paints Against Turkestan Cockroaches' provides a simple assessment of the effectiveness of three insecticide paint preparations on selected nymphal stages of Turkestan cockroach.

In my opinion, the paper is substandard in terms of today's scientific work requirements. In addition to methodological ambiguities (see Lines 11-113: 'groups of cockroaches' are mentioned instead of the number of tested individuals), it appears that the work was written without a thorough understanding of the relevant literature.

Selected critical  comments:

Typos

e.g.  Line 317 the benefits of the use of IPs in cockrorach management programs will..  

should be – “cockroach” management

Title

 “ Comparative Efficacy of Pyrethroid-Based Paints Against Turkestan Cockroaches”

The title is misleading – not all developmental stages were tested, only the nymphs of higher nymphal stages of this cockroach species.

Introduction

Line 58 -59 Insecticide paint (IP) formulations are a recent advance in enhancing the insecticidal  activity of several  against arthropod pests [13].

--Insecticidal paintings and lacquers are no recent advance (i.e. new formulations) used on cockroaches. On the contrary, they belong to the oldest formulations for cockroach control (see e.g. Cornwell 1968).

--The introduction and discussion completely lack relevant literature, both in the introduction and in relation to paintings and lacquers concerning cockroaches. This applies also to encapsulated insecticides in relation to cockroaches. Inexplicably, the authors discuss the effectiveness of paints almost exclusively in relation to mosquitoes, not cockroaches. This indicates a lack of knowledge of Cornwell  (1968), where there are numerous citations related to insecticidal coatings (The Cockroach. Volume 1. A Laboratory Insect and an Industrial Pest. Cornwell, P.B. Published by 1st. Ed. Pub. Hutchinson. 1968). I don't quite understand how it was possible to conduct experiments without taking these literary sources into account?

The methodology is substandard

Lines   94 – 97 “The IP were applied following application rates recommended by the manufacturers: 175 g/m2 on concrete and polyvinyl chloride (PVC), and 140 g/m2 on metal. Concrete received two  paint layers, according to the manufacturers’ instructions.

--It is not clear how many milligrams of the active substance were applied per square meter. Is it not expected that the reader will calculate it themselves?

Line 108 -109  The  painted substrates were allowed to dry for 7 days (according to the manufacturers’ recommendations)

 --Am I to understand that in practice, it is recommended to apply a painting insecticide to surfaces, then prevent access for pests for 7 days, let it dry, and only then make it accessible to pests? Is this really meant seriously?

Line 93- 94. Three paints were evaluated: INESFLY 7A DELTA (deltamethrin 0.25%), INESFLY 93 VESTA paint (transfluthrin 0.5%), and A-cypermethrin IP (A-cypermethrin 0.7%)

--The manufacturers and countries of origin of the products are not specified.

P. 111-113 Groups of ten third instar nymphs were placed on each substrate 112 for 15 min or 1 h.  ….  . Each treatment was 117 replicated five times.

Groups? ... The number of tested individuals is not specified!"

Figure 1.

What does this picture say, and is it not possible to understand from the description?

Results

---Why are results provided only in figures and graphs ….. any single  table can be found.

Line 164 ….The effectiveness of IPs varied according to exposure time and substrate (Fig. 3). 15  min exposures to fresh paints on concrete, metal, or PVC caused minimal mortality (≥ 86%  survivorship) (Fig. 3), which did not significantly differ from the mortality observed after  exposure to the unpainted control substrates (p > 0.05). For all substrates

---Substrates ?   Should  not be better “surfaces”?

Discussion

 Lines 271 -243   The lack of efficacy 241 of transfluthrin paints in our study contradicts reports of their high and long efficacy 242 against Anopheles gambiae mosquitoes exposed to the same formulation for similar times 243 [21].

--Contradictory? The identical preparation commonly exhibits different efficacy against various insect species from different orders. What is contradictory about that?"

Lines 343 -43:    The development of insecticide resistance in pest populations is a concern with the 339 use of pyrethroid-based paints. Outdoor painted applications are subject to a variety of 340 physical surface conditions and abiotic factors that can decrease insecticide efficacy 341 overtime [33]. Therefore, continuous exposure to sublethal doses of insecticides could 342 select resistant individuals [35]. Studies on the shelf life of insecticide-treated paints on 343 surfaces would help determine reapplication times for these materials

--The conclusions regarding resistance management are illogical. The authors state that prolonged exposure to a single type of insecticide is dangerous for resistance selection, so why even apply the same substance repeatedly ('reapplication times')?

Author Response

Response to reviewer 1

Typos

e.g.  Line 316 the benefits of the use of IPs in cockrorach management programs will..  

should be – “cockroach” management

Answer: fixed in line 308

Title

 “Comparative Efficacy of Pyrethroid-Based Paints Against Turkestan Cockroach”

The title is misleading – not all developmental stages were tested, only the nymphs of higher nymphal stages of this cockroach species.

Answer: We prefer to keep the title as it is; both abstracts specify that we used nymphs (lines 15 and 26). We also include in the keyword “nymphs”  (line 33)

Introduction

Line 58 -59 Insecticide paint (IP) formulations are a recent advance in enhancing the insecticidal  activity of several  against arthropod pests [13].

--Insecticidal paintings and lacquers are no recent advance (i.e. new formulations) used on cockroaches. On the contrary, they belong to the oldest formulations for cockroach control (see e.g. Cornwell 1968).

--The introduction and discussion completely lack relevant literature, both in the introduction and in relation to paintings and lacquers concerning cockroaches. This applies also to encapsulated insecticides in relation to cockroaches. Inexplicably, the authors discuss the effectiveness of paints almost exclusively in relation to mosquitoes, not cockroaches. This indicates a lack of knowledge of Cornwell  (1968), where there are numerous citations related to insecticidal coatings (The Cockroach. Volume 1. A Laboratory Insect and an Industrial Pest. Cornwell, P.B. Published by 1st. Ed. Pub. Hutchinson. 1968). I don't quite understand how it was possible to conduct experiments without taking these literary sources into account?

Answers: thanks for the comment which will improve the manuscript. We were unclear when writing about the novelty of paints. We meant by novel “water-based paints” that use latex, polyvinyl acetates or monomers” which is a technology that is constantly evolving. We clarify this on the text. We also revised the literature on insecticide paints as the reviewer suggested, and added in lines 60-71 and in the discussion in lines 259-263. All new references were added and reference numeration rearranged.

“Surface coatings with insecticides has been a method to improve the performance of residual insecticides [16-18]. Lacquers are common additives to insecticides used to prolong their activity against cockroaches [12]. Enhanced residual toxicity of lac-quer-formulated organophosphates [19], organochlorine [16], and pyrethroids [17] was demonstrated against German cockroaches. Micro-encapsulation of active ingredients in polymeric matrix is another development of a controlled released delivery system of insecticides reported to have residual efficacy against cockroaches and some crawling insects [12, 20]. The novel advancement of paint formulation by imbedding microen-capsulated insecticide into latex paint offers prolonged residual effect as compared to earlier insecticide paint (IP) products [21,22]. In these paints, the AIs slowly migrate to the substrate surface as the paint dries which make the insecticide available for a pro-longed time. In addition, they are ready-to-used products that can applied to surfaces by spraying or brushing [21]”.

===========

“Aging the alphacypermethrin-painted surfaces for 1 month on all substrates further increased cockroach mortality, thus indicating greater bioavailability of the AI on the surface over time. Similar results were observed in German cockroaches exposed to surfaces painted with lacquer-formulated fenvalerate aged for 90 days [17] and this was attributed to the migration of the AI to the surface”

The methodology is substandard

Lines   94 – 97 “The IP were applied following application rates recommended by the manufacturers: 175 g/m2 on concrete and polyvinyl chloride (PVC), and 140 g/m2 on metal. Concrete received two  paint layers, according to the manufacturers’ instructions.

--It is not clear how many milligrams of the active substance were applied per square meter. Is it not expected that the reader will calculate it themselves?

Answer: We created Table 1 in line 123, which has information about the treated area, total amount of applied paints and quantities of active ingredient in mg.

Also we calculated the mg of insecticide paints applied to the dark areas of the Ebeling box. This information is in lines 154-155.

Line 108 -109  The  painted substrates were allowed to dry for 7 days (according to the manufacturers’ recommendations)

 --Am I to understand that in practice, it is recommended to apply a painting insecticide to surfaces, then prevent access for pests for 7 days, let it dry, and only then make it accessible to pests? Is this really meant seriously?

Answer: This is a study under controlled laboratory conditions in which we tried to guarantee optimal conditions for evaluating the potential of the paints. Certainly, performance of insecticide paints in the field might vary according to the drying time, something that should be considered in future field studies.   

Line 93- 94. Three paints were evaluated: INESFLY 7A DELTA (deltamethrin 0.25%), INESFLY 93 VESTA paint (transfluthrin 0.5%), and A-cypermethrin IP (A-cypermethrin 0.7%)

--The manufacturers and countries of origin of the products are not specified.

Answer: information added in line 99-100

  1. 111-113 Groups of ten third instar nymphs were placed on each substrate 112 for 15 min or 1 h.  ….  . Each treatment was 117 replicated five times.

Groups? ... The number of tested individuals is not specified!"

Answer: we used groups of ten individuals each time, and run 5 replicates. We have add this information in line 126-131. Now reads:

“For each of the four IP treatments on a single substrate, ten cockroach nymphs were forcibly exposed and this experimental setup was replicated five times. This was done for both 15 min and 1 h forced exposures resulting in 1200 total Turkestan nymphs be-ing used in this study”

Figure 1.

What does this picture say, and is it not possible to understand from the description?

Answer: We improve the description of Fig. 1, now it reads in lines 136-137:

Figure 1. Groups of 10 cockroach nymphs forcibly exposed for 15 min or 1 h to surfaces treated with insecticide-based paints. PVC rings prevented cockroach escape. 

Results

---Why are results provided only in figures and graphs ….. any single  table can be found.

Answer: We added a supplemental Table, TS1 (Lines 354) which include data on cockroach mortality by day 14.    

Line 164 ….The effectiveness of IPs varied according to exposure time and substrate (Fig. 3). 15  min exposures to fresh paints on concrete, metal, or PVC caused minimal mortality (≥ 86%  survivorship) (Fig. 3), which did not significantly differ from the mortality observed after  exposure to the unpainted control substrates (p > 0.05). For all substrates

---Substrates ?   Should  not be better “surfaces”?

Answer: corrected in line 176

Discussion

 Lines 271 -243   The lack of efficacy 241 of transfluthrin paints in our study contradicts reports of their high and long efficacy 242 against Anopheles gambiae mosquitoes exposed to the same formulation for similar times 243 [21].

--Contradictory? The identical preparation commonly exhibits different efficacy against various insect species from different orders. What is contradictory about that?"

Answer: we added the word “differs” in line 242

Lines 343 -43:    The development of insecticide resistance in pest populations is a concern with the 339 use of pyrethroid-based paints. Outdoor painted applications are subject to a variety of 340 physical surface conditions and abiotic factors that can decrease insecticide efficacy 341 overtime [33]. Therefore, continuous exposure to sublethal doses of insecticides could 342 select resistant individuals [35]. Studies on the shelf life of insecticide-treated paints on 343 surfaces would help determine reapplication times for these materials

--The conclusions regarding resistance management are illogical. The authors state that prolonged exposure to a single type of insecticide is dangerous for resistance selection, so why even apply the same substance repeatedly ('reapplication times')?

Answer: We are making the text clearer so that it does not imply that we are recommending reapplying the paint with the same active ingredient.  We are modifying the text in lines 335-337:

“Studies on efficacy and shelf life of insecticide-treated paints on surfaces would help determine reapplication times and the selection of active ingredients for rotating these materials”

As the reviewer noted, continuous use of insecticide paint can lead to resistance and use of different classes of insecticides, and synergists, are strategies to delay insecticide resistance evolution. This was stated in lines 339-342

Reviewer 2 Report

Comments and Suggestions for Authors

This study evaluated the efficacy of pyrethroid-based paints for controlling Turkestan cockroaches. The manuscript presents some novel information and is in fairly good shape. However, the manuscript is poorly written and the writing throughout the manuscript needs to be improved. I would suggest the authors have the manuscript checked by a highly proficient English speaker. My recommendation is major revision.

Specific comments:

L39: do not cite websites, replace with journal publication if possible or delete.

L102: it’s not clear what “8-cm diameter circles were demarcated” means.

L 167-169: it’s not clear why paints aged for 1 month are more effective than freshly applied paints. This needs to be discussed in the results/discussion.

L224: it is unnecessary to start with “here”.

L 224: change “repellency activity” to “repellency”

Comments on the Quality of English Language

Writing throughout the manuscript needs to be improved.

Author Response

Responses to reviewer #2

This study evaluated the efficacy of pyrethroid-based paints for controlling Turkestan cockroaches. The manuscript presents some novel information and is in fairly good shape. However, the manuscript is poorly written and the writing throughout the manuscript needs to be improved. I would suggest the authors have the manuscript checked by a highly proficient English speaker. My recommendation is major revision.

 Answer: We were surprised by the reviewer’s comment. The manuscript was revised by a certified professional manuscript editor. We wish the reviewer were more specific about the segment of the text that needs improvement.  

Specific comments:

L39: do not cite websites, replace with journal publication if possible or delete.

Answer: There are few regularly updated sources of information that provide information on the spread of Turkestan cockroaches. iNaturalist is becoming ever more common in its practical usage and was created as part of an initiative with the National Geographic Society in 2017.  The only other database similar would be the Global Biodiversity Information Facility (GBIF) which shows a very similar pattern of distribution. We prefer to keep it.

L102: it’s not clear what “8-cm diameter circles were demarcated” means.

Answer: we added in line 109-11- “with pencil” to make the sentence clearer

L 167-169: it’s not clear why paints aged for 1 month are more effective than freshly applied paints. This needs to be discussed in the results/discussion.

Answer: we discussed this on lines 259-263.

L224: it is unnecessary to start with “here”.

Answer: we removed “Here” in line 224

L 224: change “repellency activity” to “repellency”

Answer: we removed “repellency”  in line 224

Reviewer 3 Report

Comments and Suggestions for Authors

This paper investigates the efficacy and repellency of using three pyrethroid-based paints against Turkestan cockroaches. The experiments are well-designed, and the manuscript is well-written. The results are properly stated, and both the introduction and discussion provide sufficient information. I recommend accepting this manuscript with a few minor revisions:

Line 68: Please add a reference after "chikungunya."

Section 2.3: It is evident from Figure 3 that the authors have included controls for this assay; however, there are no descriptions of these controls within the section. Please provide these details.

Figure 2: Could you clarify what the "A" in the left corner represents?

Line 256: Please check the spelling of "alphacypermethrin" here.

Discussion: It would be beneficial to include a discussion on the effects of insecticidal paints on non-target organisms.

Author Response

Response to reviewer # 3

This paper investigates the efficacy and repellency of using three pyrethroid-based paints against Turkestan cockroaches. The experiments are well-designed, and the manuscript is well-written. The results are properly stated, and both the introduction and discussion provide sufficient information. I recommend accepting this manuscript with a few minor revisions:

Line 68: Please add a reference after "chikungunya."

Answer: we added the reference 21 to line 77. Complete reference is in lines 424-426:

Inesfly Corporation. Control of Aedes genus mosquitoes. Dengue, Yellow fever, Chikungunya, Zika. Valencia, Spain. https://inesfly.com/wp-content/uploads/2021/08/INESFLY-SOLUTIONS-FOR-AEDES.pdf. Accessed February 2024.

Section 2.3: It is evident from Figure 3 that the authors have included controls for this assay; however, there are no descriptions of these controls within the section. Please provide these details.

Answer: we added this information in the new supplementary table (Table S1, line 354)

Figure 2: Could you clarify what the "A" in the left corner represents?

Answer: it should not be there. It has been removed from Fig. 2

Line 256: Please check the spelling of "alphacypermethrin" here.

Answer: corrected in 256

Discussion: It would be beneficial to include a discussion on the effects of insecticidal paints on non-target organisms.

Answer: we have added in lines 328-332.

“Insecticide-based paints might impact beneficial insects resting or nesting in areas where these paints are intended to be used. Where there is a concern on this impact, paints should be restricted to areas less likely to be occupied by beneficial insects such as crawl spaces and attics.”

Reviewer 4 Report

Comments and Suggestions for Authors

A relevant and timely article, written and presented well. I enjoyed reading it!

Introduction, methods, results are very good. Discussion section could be made more concise. 

Potential concerns about the technology are effects on non-targets coming in contact with the painted surfaces, when the technology is used in the field. E.g., isopods, honey bees nesting in irrigation boxes, ground-dwelling bees, etc. There is possibility for prolonged exposure of these non-targets to the paints.  If literature is available, this could be mentioned in the discussion section.

Please check the References section for formatting.

Author Response

Response to reviewer # 4

A relevant and timely article, written and presented well. I enjoyed reading it!

Introduction, methods, results are very good. Discussion section could be made more concise. 

Potential concerns about the technology are effects on non-targets coming in contact with the painted surfaces, when the technology is used in the field. E.g., isopods, honey bees nesting in irrigation boxes, ground-dwelling bees, etc. There is possibility for prolonged exposure of these non-targets to the paints.  If literature is available, this could be mentioned in the discussion section.

Answer: we addressed in lines 328-332

Please check the References section for formatting.

Answer: references checked

Reviewer 5 Report

Comments and Suggestions for Authors

This is a valuable and well written paper.  I only have two comments for improvement.

First, the PI formula (line 153), while simple enough is not entirely obvious to someone who doesn't use it routinely.  Is it correctly stated as 1 - quotient shown in brackets, or is it 100 - quotient?  A little more explanation and a simple example calculation would be helpful.

Secondly, authors don't really discuss the possible mechanisms for the time effect on mortality caused by the paints.  In lines 170-172, why was mortality higher in paint aged for 1 month compared to freshly-applied paints? 

Comments on the Quality of English Language

Very minor edits are needed.

Author Response

Response to reviewer # 5

 This is a valuable and well written paper.  I only have two comments for improvement.

First, the PI formula (line 153), while simple enough is not entirely obvious to someone who doesn't use it routinely.  Is it correctly stated as 1 - quotient shown in brackets, or is it 100 - quotient?  A little more explanation and a simple example calculation would be helpful.

Answer: Based on recommendation of another reviewer, we removed data and graph associated with PI formula.

Secondly, authors don't really discuss the possible mechanisms for the time effect on mortality caused by the paints.  In lines 170-172, why was mortality higher in paint aged for 1 month compared to freshly-applied paints? 

Answer: we discussed this in lines 259 and 263.

Reviewer 6 Report

Comments and Suggestions for Authors

Please review my comments within the attached annotated pdf. Some references to be cited have been suggested. Notably, the taxonomic nomenclature of the species has changed recently. Also, authors treat liquid insecticide sprays as the "standard" control measure even though no field-based studies demonstrate efficacy. Conversely, recent publications have reported on the efficacy of granular and gel bait formulations. Why the focus on sprays?

Author Response

Response to reviewer # 6

Please review my comments within the attached annotated pdf. Some references to be cited have been suggested. Notably, the taxonomic nomenclature of the species has changed recently. Also, authors treat liquid insecticide sprays as the "standard" control measure even though no field-based studies demonstrate efficacy. Conversely, recent publications have reported on the efficacy of granular and gel bait formulations. Why the focus on sprays?

The word "relies" suggests that sprays are effective and common. Though research may be lacking, I'm not sure I would call sprays effective for this pest, regardless of the active ingredient used. There are several reasons for this. Baits, on the other hand, have been shown as effective (see Sutherland et al, 2022 in suggested references) and are increasingly common within professional control protocols” (line 9-10)

Answer: we have modified the text according to the reviewer’s recommendation in lines 9-10:

“The application of pesticide sprays in and around areas where cockroaches aggregate is common practice in trying to reduce their populations”

The entomological nomenclature for this pest has recently been changed (from Shelfordella lateralis) to Periplaneta lateralis Walker (see Deng et al, 2022 and Luo et al 2023 in suggested references (line 19).

Answer: We replaced “Blatta” for “Periplaneta” in line 37

See above: does "standard" mean effective? Liquid sprays may be common, but they may not be effective. Perhaps calling them "most common", "conventional", or "prevailing" would be more correct. Also, baits are increasing in use and have proven efficacy (line 21)

Answer: we modified the text according to the reviewer’s recommendation in lines 20-22.

Treatments with liquid spray formulations containing insecticides is the most conventional method to decrease Turkestan cockroach population abundance around buildings

What about repellency?(line 30)

Answer: We added the word “repellency” (line 30-31)

The reviewer suggested the word “on” instead of “in” (line 31)

Periplaneta instead of “Blatta” (line 37)

“Other regions” instead of “various parts” (line 40)

Entomological nomenclature required here: Blatta orientalis...(line 41)

Answer: “Blatta orientalis” was added in lines 42-43

"in areas around homes and other structures" as suggested by the reviewer in line 44

These references do not include efficacy data against this species. To my knowledge, the only field data published on control of this species was by Sutherland et al, 2022 (see suggested references to add). This sentence needs to be reworded or rewritten. Sprays are not very effective, in my opinion.

Answer: we modified the text in lines 48-50 following reviewer’s suggestion:

“For the management of this cockroach, programs that include outdoor application of baits and exclusion have demonstrated to reduce populations within a month [9]. Reference were added in the reference list.

This is referring to a lab study. Suggest "have been shown to be effective in laboratory studies" or "may be effective" (line 48)

Answer: we modified the text, lines 50-51:

“Although common insecticide formulations have shown to be effective against Turkestan cockroaches in laboratory studies”

This was already defined above. No need to do it again.

Answer: This sentence was modified and correction was not needed

Toxicity tests were performed with forced exposures on common 73 substrates used by these species around peridomestic areas. Repellency, toxicity, and po-74 tential for effectiveness of IPs against these cockroaches in field conditions were deter-75 mined with Ebeling choice box experiments.

This information belongs in Methods section, not in Intro.

Answer: we delete this sentence from here.

What was the product name?

Answer: names was added lines 105, 107

 More detail required here: were the insects able to climb the walls of the 8 cm pipe to avoid exposure to the treated surface? (113-114)

Answer: we modified the text following reviewer’s recommendations (lines 131-132):

This needs to be added above, where I have asked the question about confinement. A citation, if it exists, would be best to establish this behavioral fact.

Answer: It was moved as suggested and we added (A.R., pers. comm.) (line 132).

Authors report that only half of the "dark side" was treated. Were insects harboring on the non-treated portion of the dark side repelled? They were not counted as such. This could reveal contact repellency / irritability and should be reported.

Answer: The description in the text is incorrect: we meant “half of the box” and not “half of the dark side”.  Th word half was deleted.

This equation does not easily parse repellency from toxicity. It may be most illuminating to report simplified data showing these effects singly. Your results show, for instance, that repellency may have decreased overall efficacy (line 152)

Answer: This is an excellent advice from the reviewer and make the manuscript clearer. We opted to present repellency data and mortality only, and remove text and graphs associated with the performance index.     

“Forces” was replaced by “forced” (line 227)

"encapsulated"?

Answer: it is, correction was made in line 233

“Such” replaced by “that” (correction in line 233)

 "encapsulated"?

Answer: it is, correction was made (line 238)

This is why authors should have checked the location of insects harboring in the dark portion of the boxes but on untreated surfaces. Also, I think cypermethrin is considered somewhat volatile. Could mortality be due to volatile action? Authors should examine the literature on volatility of pyrethroids and cypermethrin specifically.

Answer: Please see below the addition (lines 257-258):

Also, vapors released from the insecticide-painted surfaces could also contributed to cockroach mortality [33]). Reference added to the reference list.

This paper would be much stronger if authors observed effects over longer time periods, especially given that efficacy appeared to increase as paint dried.

Answer: we agree with the reviewer. The literature reports a better performance as the paints dry. We are recommending that future studies evaluate toxicity over longer time periods. We added the following in lines 313 - 314:

“and future studies should evaluate the efficacy of these materials for longer periods”.

 Furthermore, it's worth noting that pyrethroid resistance has been documented in Periplaneta americana, which is the sister taxon according to Luo et al.

Answer: We agree, there is a potential for development of insecticide resistance in cockroaches.  We added the words “Periplaneta sps” and added reference 43 to the reference list.  

Round 2

Reviewer 2 Report

Comments and Suggestions for Authors

Good job on the revision.

Author Response

No changes were required by this reviewer